



# Technical Note: Noble gas extraction procedure and performance of the
Cologne Helix MC Plus multi-collector noble gas mass spectrometer for
cosmogenic neon isotope analysis
Benedikt Ritter[1*], Andreas Vogt[1], Tibor J. Dunai[1*]
[1] University of Cologne, Institute of Geology and Mineralogy, Zülpicher Straße 49b, Köln 50674,
Germany
*Corresponding authors
Benedikt Ritter – benedikt.ritter@uni-koeln.de
Tibor J. Dunai – tdunai@uni-koeln.de
Keywords: Noble Gas, Mass Spectrometry, Cosmogenic Nuclides
**Abstract:**
We established a new laboratory for noble gas mass spectrometry that is dedicated for the
development and application to cosmogenic nuclides at the University of Cologne (Germany). At
the core of the laboratory are a state-of-the-art high mass resolution multicollector Helix MCPlus
(Thermo-Fisher) noble gas mass spectrometer and a novel custom-designed automated
extraction line. The Mass-spectrometer is equipped with five combined Faraday Multiplier
collectors, with $10^{12}\,\Omega$ and $10^{13}\,\Omega$ pre-amplifiers for faraday collectors. We describe the extraction
line and the automized operation procedure for cosmogenic neon and the current performance of
the experimental setup. Performance tests were conducted using gas of atmospheric isotopic
composition (our primary standard gas); as well as CREU-1 intercomparison material, containing
a mixture of neon of atmospheric and cosmogenic composition. We use the results from repeated
analysis of CREU-1 to assess the performance of the current experimental setup at Cologne. The
precision in determining the abundance of cosmogenic $^{21}$Ne is equal or better than those reported
for other laboratories. The absolute value we obtain for the concentration of cosmogenic $^{21}$Ne in
CREU is indistinguishable from the published value.
## 1. Introduction
Cosmogenic Ne isotopes are stable and compared to other cosmogenic radionuclides (e.g. $^{10}$Be,
$^{26}$Al) exhibit the potential to date beyond the physical limit of radionuclides. The particular
strength of cosmogenic neon is its application to date quartz clasts of very old surfaces (>4Ma) or
very slowly eroding landscapes (<10cm/Ma), which is unattainable with most other radionuclides
(Dunai, 2010). Cosmogenic Ne analysis can be applied to a range of neon-retentive minerals (e.g.,
quartz, olivine and pyroxene); amongst which quartz is the most commonly used. Ne can be
measured on conventional sector field noble gas mass spectrometers; is less time consuming and
requires less sample-preparation compared to AMS measurements required for the cosmogenic





radionuclides. Recent studies used cosmogenic Ne isotope geochronology for dating old surfaces
(e.g. Ritter et al., 2018; Dunai et al., 2005; Binnie et al., 2020), reconstructing erosion rates (e.g.
Ma et al., 2016), or applying $^{10}Be/^{21}Ne$ burial dating (e.g. Mcphillips et al., 2016). The advantage
to use also other minerals than quartz, led to several studies using $^{21}Ne$ to date for example basalt
flows (e.g. Espanon et al., 2014; Gillen et al., 2010). Neon has three stable isotopes $^{20}Ne$, $^{21}Ne$, and
$^{22}Ne$, of which $^{20}Ne$ is the most abundant; the atmospheric $^{21}Ne/^{20}Ne$ and $^{22}Ne/^{20}Ne$ ratios are
0.002959 ± 0.000022 and 0.1020 ± 0.0008, respectively (Eberhardt et al., 1965). There are several
recent re-determinations of the atmospheric $^{21}Ne/^{20}Ne$ ratio (e.g. Honda et al., 2015; Wielandt and
Storey, 2019) one of which yields a ~2% lower value (Honda et al., 2015). For our evaluation of
our data, we utilize the $^{21}Ne/^{20}Ne$ value of Wielandt and Storey (2019) of 0.0029577 ± 0.0000014
and for $^{22}Ne/^{20}Ne$ that of Eberhardt et al. (1965). Note, that in the context of the determination of
the *abundance* of cosmogenic nuclides in a sample eventual differences between the used and the
actual value of the atmospheric $^{21}Ne/^{20}Ne$ ratio are unimportant, if (i) atmospheric neon is used
as calibration gas, (ii) the same value for the atmospheric composition of atmospheric neon is
used consistently throughout the evaluation of the isotope data (mass discrimination etc.) and
calculation of abundances and (iii) the atmospheric value used is reported along with the data.
All three neon isotopes are produced in about equal proportions by neutron spallation in quartz
(Niedermann et al., 1994). Due to the lower abundances of $^{21}Ne$ and $^{22}Ne$ as compared to $^{20}Ne$ in
air, and the ubiquitous presence of atmospheric neon in samples, any contribution from
cosmogenic production in samples is most easily picked up with the former two isotopes.
Consequently, the neon three-isotope diagram with $^{20}Ne$ as common denominator (Niedermann
et al., 1994; Niedermann, 2002) is customarily used to asses $^{21}Ne$-data for the presence of
terrestrial cosmogenic Ne and its discrimination from other non-atmospheric Ne-components
(Dunai, 2010). The latter may be nucleonic Ne and/or mantle-derived Ne. Hence, the accurate
determination of cosmogenic Ne and its discrimination from other components requires the
accurate discrimination from any other component.
Common isobaric interferences for neon measurements are at m/e =20 $^{40}Ar^{2+}$, $H^{19}F^+$ and $H_2^{18}O^+$
interfering with $^{20}Ne^+$, and at m/e= 21 $^{20}NeH^+$, interfering with $^{21}Ne^+$, and $^{44}CO_2^{2+}$ at m/e=22
interfering with $^{22}Ne^+$. $^{40}Ar^{2+}$ and $^{12}C^{16}O_2^{2+}$ interferences are considered to be the main challenges
for neon analysis. Recent studies demonstrated the ability of the Helix MCPlus to fully resolve the
$^{40}Ar^{2+}$, $H^{19}F^+$ and $H_2^{18}O^+$ peaks from the $^{20}Ne^+$ peak (e.g. Honda et al., 2015; Wielandt and Storey,
2019) and its ability to reliably measure $^{21}Ne$ at an off-centre peak position that is free of
interference from $^{20}NeH^+$ (Honda et al., 2015; Wielandt and Storey, 2019). The remaining
interference of $^{12}C^{16}O_2^{2+}$ at m/e=22 can be corrected via monitoring of the double/single-charged
ratio of $CO_2$ in-between samples (Honda et al., 2015) or the measurement of $^{13}C^{16}O_2^{2+}$ at m/e=
22.5 during sample analysis (Wielandt and Storey, 2019). Recently mass spectrometers with



higher resolution have become available, which permit almost full separation of $^{12}C^{16}O_2^{2+}$ and $^{22}Ne$
(Farley et al., 2020).
Beside the resolution and characteristics of a noble gas mass spectrometer to resolve and
quantitatively determine neon compositions of an unknown sample, the calibration, sample
extraction and purification are crucial achieving accurate and reproducible results. Automation of
extraction protocols and workflows may assist in achieving a high degree of reproducibility by
eliminating inaccuracies or errors by operators having a variable degree of expertise. In this
paper, we describe the current setup of the noble gas mass spectrometer and its automated
extraction line that is located in the Institute of Geology and Mineralogy at the University of
Cologne (Germany), and we review its performance for neon analysis.

## 2. Experimental setup

### 2.1 Noble gas mass-spectrometer

The Cologne noble gas laboratory is equipped with a Helix MCPlus from Thermo Fisher Scientific
with five CFMs modules (Combined Faraday Multiplier), called 'Aura'. The mass spectrometer
configuration and performance is mostly equivalent to those described elsewhere (Honda et al.,
2015; Wielandt and Storey, 2019); here we describe potential differences in configuration and
performance parameters that may be unique to a given instrument.
In the instrument at Cologne University, all but one Faraday amplifier, are equipped with $10^{13}\,\Omega$
resistors, one with $10^{12}\,\Omega$ (H2). The L1 module has 0.3 mm wide collector slits, all other modules
have 0.6 mm wide slits. The CFM at L1 configuration is flipped (i.e., the relative positions of the
Faraday and Multiplier are swapped) as compared to the standard configuration, which is the only
difference from the standard configuration. The two SAES NP10 getters, at the source and the
multiplier block, are kept at room temperature during analysis.
For neon isotope analysis of calibrations and samples, we utilize the H1, Ax and L1 CFMs ($^{20}Ne^+$
L1 Faraday; $^{22}Ne^+$ H1 Faraday; $^{21}Ne^+$ L1 multiplier; $CO_2^+$ H1 Faraday; for blanks we utilize the L1
multiplier also for $^{20}Ne^+$ and $^{22}Ne^{++}$). With the widest source slit (0.25 mm) mass resolution (at 5%
peak valley) and mass resolving power (between 10% and 90% of peak) on the L1 detector with
0.3 mm collector slit width are approximately 1700 and 6500, respectively. For the Ax and H1
detectors with 0.6 mm collector slit, the corresponding values are approximately 1000 and 6000,
respectively. As such the system allows the interference-free determination of $^{20}Ne$ and $^{21}Ne$; for
$^{21}Ne$ this entails measuring at an off-centre peak position (Honda et al., 2015; Wielandt and Storey,

103   2019).



### 2.2 Extraction line

The original noble gas extraction and purification line has a modular design. Modules are (i) extraction (currently only laser extraction; to be joined by a crushing device), (ii) calibration gas pipettes and volumes, (iii) clean-up, and (iv) cryogenic separation. The calibration module is physically linked to the clean-up module, the other modules can be separated, if required. Among the common features of all modules is that, all valves and tubing in contact with the sample gas are of metal; tubing is of stainless steel or vacuum-annealed copper. Furthermore, all valves used for handling of sample and calibration gas are pneumatically actuated all-metal diaphragm valves (Fujikin MEGA-M LA; FWB(R)-71-6.35), that can be operated at high temperature (up to 350ºC). Tubing and valves in contact with sample gas are continuously kept at constant temperature between 160ºC and 200ºC; exceptions are the functional traps and portions of the tubing in the cryogenic separation. Temperature is maintained with heating tapes (Horst HS 450ºC) and is controlled section-wise (Horst HT30). The temperature of the heated sections is controlled to ±1ºC. Thermal insulation is achieved with high-temperature resistant silicone foam (HOKOSIL®; resists ≤280ºC; permitting bake-out at higher than operation temperatures). Vacuum connections used are VCR (for Fujikin Valves), CF (for adapters, getters and manifold in clean up) and Swagelok (for flexible tubing between modules and between ports of the cryogenic separator (Swagelok 321 Stainless Steel Flexible Tubing with XBA adapter; and copper tubing). Tubing and valves are 1/4" outer diameter (Swagelok) or equivalent (VCR, Fujikin). The overall internal volume of the extraction line (laser-extraction, clean-up & cryogenic separation) is 530 cm$^3$. Outside the volume used for sample preparation, CF connections are used throughout. A schematic overview and picture of the extraction line is provided in Fig. 1.

More specifically about the individual modules:

i.     Laser extraction module: Energy for the heat-extraction is provided by an output-tuneable 600 W fiberlaser (Rofin StarFiber600) at 1064nm wavelength through galvanometer scanner optics (Rofin RS S 14 163/67 0°) and a sapphire viewport (Kurt Lesker, VPZL-275DUS). Quartz samples for neon-extraction are heated in 15 mm outer diameter tungsten cups with lids. For neon-extraction of quartz, the heating occurs via scanning of the lids (scanning speed 20 cm/s; rastering a circular area of 10 mm diameter) with a defocussed (~ 0.5 mm diameter) continuous wave beam with 100W power for 15 min. Copper (melting point 1085ºC), placed in the cup-assemblies, melts at 80W laser power (15 min extraction time); we assume that at 100W laser power the internal temperature is ≥1200ºC. The temperature of the top of the tungsten lids is monitored with a pyrometer (CellaTemp PA 29 AF 2/L; Keller HCW). The tungsten cups can hold up to ~600 mg quartz, which is fully extracted at aforementioned conditions. The tungsten cups are reused. When analysing quartz, tungsten cups are emptied with a suction micropicker (Micropicker MPC100; VU Amsterdam), while remaining in the sample revolver. In cases where



samples are melted during extraction, tungsten cups could be cleaned in HF (then of course
outside the revolver). Up to eighteen tungsten cups are loaded in a sample revolver, housed in a
DN 200 CF flange-sandwich. The sample revolver is machined from molybdenum, which permits
the heating of the tungsten cups while being situated in the revolver. To minimize heat-loss
through conduction, the cups sit on shards of zirconia (synthetic, cubic-stabilized $ZrO_2$). For
sample loading the volume containing the revolver is vented and continuously flushed with high-
purity nitrogen. During laser extraction the pressure is monitored (MEAS EPB-C1 sensor, welded
into a male VCR connector; Disynet), in case of an eventual failure of the viewport, the extraction
volume is automatically purged with Ar. The laser extraction has a dedicated pumping unit
(Pfeiffer HiCube80); pressures attained after sample loading and heating of the revolver (via
short-term laser-heating – stepwise increased to 200W -of an empty tungsten cup; the external
housing flanges reach ~50ºC during this treatment; temperatures in adjacent cups in the revolver
stay below 156,6ºC, which was verified with Indium wire) are usually <$5*10^{-9}$ mbar (the lower
limit of the pressure gauge used) after one night of pumping.  Typical blanks, obtained via heating
of an empty tungsten cup assembly, are ~0.3 fmol Neon. A detailed description of this novel
laser-furnace will be provided elsewhere.
ii.      Calibration gas pipette module: The gas-pipettes are assemblies of male and female
versions of pneumatically actuated Fujikin diaphragm valves (MEGA-M LA; FWB(R)-71-6.35); the
reservoirs were manufactured by Caburn-MDC, the insides of the reservoirs are electropolished.
We currently have three different gases available for noble gas calibration ('Linde', 'Air', 'RedAir').
'Linde' is a noble gas mixture in nitrogen (9.889±0.009% He, 10.00±0.01% Ne, 10.01±0.01% Ar,
0.00987±0.0003% Kr; 0.01023±0.00002% Xe; all uncertainties are ±2σ; remainder $N_2$; prepared
gravimetrically by Linde) the He is enriched in $^3$He (12.3±0.3 $R_a$; ±2σ), the remaining noble gases
have atmospheric composition.  'Air' is a reservoir of air at atmospheric pressure and 'RedAir' a
reservoir of air at reduced pressure. For the neon determinations we utilize 'RedAir'. The volumes
of all reservoirs and the pipettes have been determined using a gravimetrically calibrated gas
volume (an assembly of a Swagelok SS-4H valve and a Swagelok SS-4CS-TW-50 miniature
cylinder; repeatedly weighed (Satorius MSA524P-1000-DI) under vacuum and filled with air at a
temperature, pressure and relative humidity measured with traceable and/or certified sensors
(thermometer: testo 110; manometer: Greisinger GMH 3181-12, DKD certificate D19853, D-K-
15070-01-01; hygrometer: VWR traceable 628-0031) and pressure readings (MKS Baratron, Type
628FU5TCF1B) from repeated step-wise expansion of gasses into the pipette-reservoir
assemblies). The temperature in the room where these calibrations were conducted was stable to
±0.5ºC over the course of the calibrations. The volumes of the reservoir and pipette of 'RedAir' are
8740±35 $cm^3$ and 1.4565±0.0006 $cm^3$, respectively. For filling of the 'RedAir' reservoir one pipette
volume of air was expanded into the reservoir; the temperature, pressure and humidity at the




time of filling of the pipette were measured with a traceable and certified sensor (same as above).
The first pipette volume extracted from the 'RedAir' reservoir contained $4.0196\pm0.0027 \times 10^{-9}$ cm$^3$
atmospheric neon at standard temperature and pressure ($179\pm1$ fmol atmospheric neon).
iii.    Clean-up module ('Sputnik'): Arranged around a central hexagonal 8-port manifold
(Kimball Physics, 2.75" spherical hexagon) are the sample/calibration inlet, the pumping outlet, a
pipette leading to a residual gas analyser (Hiden HAL/3F PIC), two SAES NP50 getters (one
operated hot, the other at room temperature; getters are housed in SAES GP 50 W2F bodies; water
cooling is optional, not used during sample analysis), an optional expansion volume, an internally
heated capacitance manometer (MKS Baratron, Type 628FU5TCF1B; @ 100ºC) and the outlet to
the cryogenic separation unit (Fig. 1). The sample/calibration inlet tubing has an auxiliary port,
which e.g., is used for the crushing extraction module (build around a T4S crushing unit, VU
Amsterdam). The clean-up module is pumped via a manifold connected through gate-valves (MDC
E-GV-1500M-P) to a turbopump (Pfeiffer HiPace 300; backed by a membrane pump, Pfeiffer MVP
030-3) and an iongetter pump (Agilent, Vacion 40 plus Starcell).
iv.    Cryogenic separation module: Centre of this module is a double-cold trap unit (Janis, twin
coldhead model 204) that has inlet and outlet lines to three traps: a watertrap (operated at 205K)
a bare steel trap ($\geq$24K) and a charcoal trap ($\geq$ 10K). The cold trap unit is controlled by a
Lakeshore 336 Controller (Cryotronics). This module is pumped by an ion pump (Agilent, Vacion
40 plus Starcell).
The performance of the bare cold trap unit for He, Ne, Ar-separation was calibrated using the
Residual Gas Analyser (Hiden HAL/3F PIC). Neon is quantitatively adsorbed on the bare trap at
24 K, in equilibrium about 60% of the helium is adsorbed at 24 K. We use this to separate helium
from neon. Helium is removed (distilled-off in disequilibrium) either to the ion-pump or the 10K
charcoal head, the latter if the He is to be retained for analysis. Neon is fully released from the bare
trap at 80K, at this temperature argon is quantitatively retained on the bare trap, permitting
quantitative separation of the two gases.
Besides its functionality to separate noble gases from each other the bare trap serves as coldtrap
during Ne-analysis (held at 80 K) and replaces a liquid nitrogen cooled trap, which would
otherwise customarily be used for this purpose. The latter may introduce intensity fluctuations
during analysis due to changing coolant level, which we avoid with our set up. The last
pneumatically actuated valve before the Helix-Plus MCMS serves as inlet valve, the manual valve
of the Helix-Plus MCMS is permanently open.





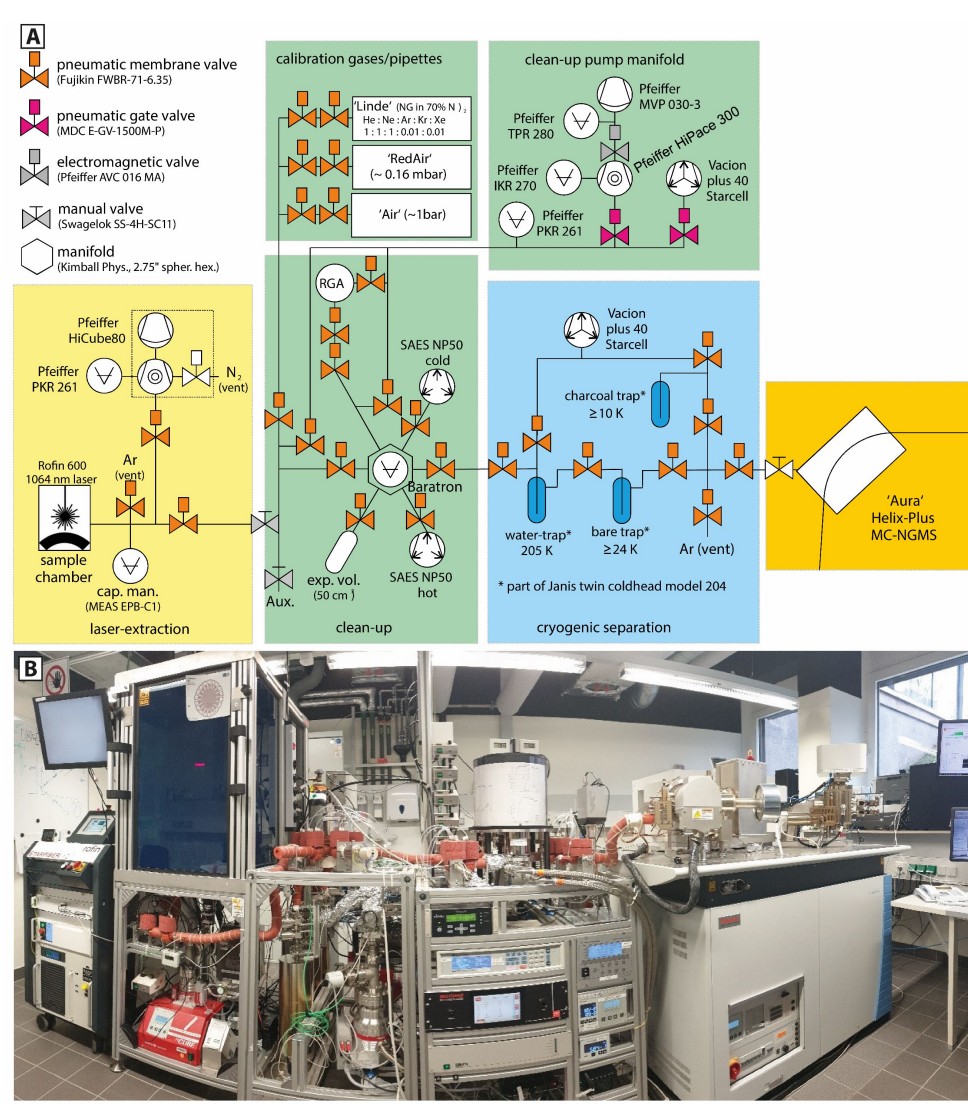


*Fig. 1: (A) Schematic plan and picture (B) of the noble gas extraction and purification line at the*
*University of Cologne. From left to right: Rofin Starfiber 600, full-protection laser-cage (laser*
*protection windows P1P10, Laservision) housing the laser extraction, clean up unit, cryogenic*
*separation unit and the Helix Plus NG-MCMS 'Aura'. The laboratory is temperature-stabilized to*
*±0.5ºC. Further description is provided in the text.*
**2.3 Automation**
The extraction and purification line can either be operated manually, via a switchboard for the
pneumatic valves and the components' original controllers, or automatically via LabView. Manual



operation is mainly used for development of analytical routines, automatic operation generally
for sample and calibration-gas analysis. Automatic operation liberates the operator from
conducting necessarily repetitive tasks, thus helps to prevent mistakes and inconsistencies from
oversight or negligence; it allows to conduct gas purification and separation under precisely
identical conditions. The latter is also assisted by avoiding liquid coolants, which commonly are
affected by variable coolant levels (unless automatically filled with a suitably precise system or an
experienced and conscientious operator). Currently the laser system is operated manually (due
to safety regulations); all subsequent steps - until admission of the gas to the mass spectrometer
- are automated utilizing LabView (Version 2018) in a Windows 10 environment. The mass
spectrometry analysis of the purified gas is conducted with Qtegra (Thermo Fisher Scientific).
Valve control electronics were developed and implemented in-house, including digital
input/output modules (I/O modules from National Instruments) and RS-232 communication.
Main devices such as, SAES getter control, Lakeshore Cryo-Controller, turbo and ion-pumps
offered already LabView compatible Sub-VI's (Virtual Instrument), which were implemented into
the operation VI. The Agilent Ion Pump Control connection via the computer interfaces were
written/developed in-house.
The gauges and controllers of the Turbo pumps (Pfeiffer) and Ion pumps (Agilent) are monitored
via the operation VI. Automatic safety protocols are implemented to protect the extraction line
and equipment against sudden pressure increases. Temperature setting and monitoring of the
three cold traps (Janis Cryostat) is performed by the Lakeshore 336 controller, which in turn is
controlled via the operation VI.
LabView computing of the extraction sequence/protocol was programmed in single commands
and steps, joined into command sequences connected in series as sub-VIs for each extraction
protocol (various noble gases and sources of samples or calibration gas). Pressure and
temperature control sequences are programmed in continuous loop to ensure stability and safety
during operation. For handling, a structured user interface was designed (Fig. 2), which provides
the user with information about all parameters, total duration, and additionally logs every
extraction step.



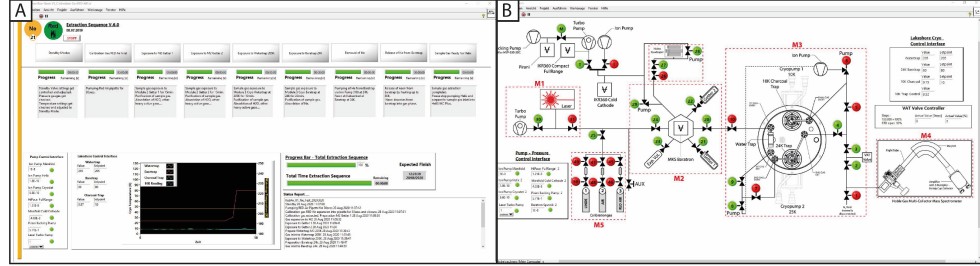


*Fig. 2: Operating VIs of the Cologne Noble Gas Helix MCPlus. (A) The Neon VI informs the user in real time about current data, such as pressure and temperature, as well as about the current status of the preparation and the different extraction steps. The process is fully automatic, the user is informed about the estimated extraction time. (B) Valve circuit overview.*

## 3. Analytical Procedure

Quartz samples are cleaned using standard procedures using dilute HF as etchant (Kohl and Nishiizumi, 1992). Up to 600 mg of quartz are loaded into tungsten-cups and covered with a tungsten-lid, the latter has a small opening in the lid to facilitate gas release. When opening the laser furnace for re-loading, the furnace is vented and purged with a continuous flow of pure nitrogen. In normal operation, after the initial installation and bake-out, the internal parts of the furnace are never again exposed to air. The tungsten cups and lids remain in the nitrogen atmosphere during sample (re-)loading. Cups are emptied with a suction micropicker (Micropicker MPC100, VU Amsterdam) while seated in the revolver, and weighed samples are transferred from glass vials through a miniature metal funnel (glass funnels produced undesirable static effects) into the cups. After reloading, the sample revolver is heated by firing the laser on an empty cup; pressure $<5\times10^{-9}$ mbar is usually achieved after pumping overnight. During this clean-up, and during subsequent analyses, the temperature of adjacent cups does not exceed 156.6ºC (verified with Indium wire). Cosmogenic Ne is extracted from quartz by heating the sample with a defocussed laser beam at 100W for 15 min; at these settings the cup-insides reach ~1200 °C. This temperature allows reliable extraction of cosmogenic neon (Vermeesch et al., 2015). After heating the furnace, it is allowed to cool for five minutes before the sample is expanded to the clean-up module.

For calibrations, the calibration gas is expanded for 30sec into the pipette, the pipette volume is then expanded into the clean-up volume. After this step, purification is identical for sample and calibration gases. The pipetting of calibration gas, and the purification of sample and calibration gases, is fully automatized.

Reactive gases are removed by sequential exposure to two metal getters (SAES NP50); the first is operated hot, the other at room temperature. The gas is exposed to each for 15 min. Subsequently



276 the gas is exposed to the water trap at 205K for 10min. The remaining inert gases are exposed to

277 the bare-metal trap at 24K for 20min, which is then pumped for 5 min to remove helium from the

278 sample gas. The trap is then isolated and heated to 80K, followed by five-minutes holding time for

279 re-equilibration. Neon is quantitatively released and argon is quantitatively retained on the trap.

280 Ensuing Ne gas is expanded into the Helix MCMS for analysis. The bare trap at 80 K remains

281 connected to the mass spectrometer during analysis, for pumping of $CO_2$ and Ar evolving from the

282 mass spectrometer.

283 The configuration of the Helix is described above. For maximum sensitivity and precision for

284 abundance determination (Wielandt and Storey, 2019), we use the widest (0.25 mm) source slit

285 for neon analysis. We run the source at an electron energy of 115 eV, trap current of 200 µA and

286 an acceleration voltage of 9.9 kV.

287 $^{20}Ne$ is measured on the high-resolution L1 Faraday cup (fitted with $10^{13}$ Ω pre-amplifier), fully

288 resolved from $^{40}Ar^{2+}$ and from molecular interferences such as $HF^+$, $H_2^{18}O^+$. $^{21}Ne$ is measured

289 off-centre on the high-resolution L1 multiplier, at a position that is free from interference from

290 $^{20}NeH^+$. $^{22}Ne$ is measured at peak centre on the H1 Faraday cup (fitted with $10^{13}$ Ω pre-amplifier);

291 interference from $CO_2^{2+}$ is corrected via monitoring of the double/single-charged ratio of $CO_2$

292 in-between samples and measurement of $CO_2$ during sample analysis, which we found to be stable

293 at 0.0437±0.001 for our system throughout the period for which the data we report here were

294 obtained. The corresponding corrections of $^{22}Ne$ intensities are < 0.3% for one shot of 'RedAir'

295 calibration gas (~17 fmol $^{22}Ne$); the uncertainties of the correction are ~2 %, i.e., add < 0.006%

296 uncertainty of the intensity determinations for 'RedAir'; these values scale linearly for smaller or

297 larger amounts of $^{22}Ne$ as found in samples. $CO_2^+$ is measured on the Faraday cup of the Axial

298 collector (fitted with $10^{13}$ Ω pre-amplifier). We refrain from analysing the larger Neon-beams

299 ($^{22}Ne$, $^{20}Ne$) on the multipliers, since we found that they are a significant source of $CO_2$ upon being

300 hit by beams larger than those typical for $^{21}Ne$ signals (for analysing blanks, however, we use a

301 multiplier for $^{20}Ne$ and $^{22}Ne$). Besides, the Faraday cups have a superior linearity and stability over

302 time (Wielandt and Storey, 2019). The mass spectrometer sensitivity, mass-discrimination and

303 multiplier vs. Faraday gain is calibrated with 'RedAir', which is measured at least once a day during

304 sample runs. Each batch of samples includes at least one measurement of ~100mg CREU-1

305 (Vermeesch et al., 2015) to monitor the performance of the extraction and purification system.

306 We are in the process of producing a new intercomparison material to replace CREU-1, whose

307 supplies are limited and eventually will run too low for regular use.

308 **4. Performance**

309 The within-run reproducibility of Neon-isotope ratios as determined for calibration gas ('RedAir',

310 ~ 17 fmol atmospheric Ne) is similar for $^{21}Ne/^{20}Ne$ and $^{22}Ne/^{20}Ne$ ratios, with 0.46% and 0.37%





(±1σ, n=52), respectively. This dispersion is larger than the uncertainty of individual
measurements (Fig. 3); this feature, and the values for dispersion, are similar to those reported
for other Helix Plus instruments (Honda et al., 2015; Wielandt and Storey, 2019). We use the
means and the uncertainty of the means of calibrations within runs to calibrate the measurements
samples, i.e., propagate the observed dispersion in calculations of the abundance of cosmogenic
$^{21}$Ne in samples. The calculated cosmogenic $^{21}$Ne abundances from 22 aliquots of CREU-1 (Table
1) all agree within 2σ with their arithmetic mean (348 ±10 x $10^6$ atoms/g; ±2σ); thus, we may
calculate an error-weighted mean: 348 ± 2 x $10^6$ atoms/g (±2σ), which is indistinguishable from
the published value (348± 10 x $10^6$ atoms/g; Vermeesch et al., 2015). We conclude that the
reproducibility and accuracy of the current set up at the University of Cologne for determining
cosmogenic $^{21}$Ne in quartz is similar to or better than those reported for other laboratories
worldwide (Vermeesch et al., 2015; Farley et al., 2020; Ma et al., 2015).

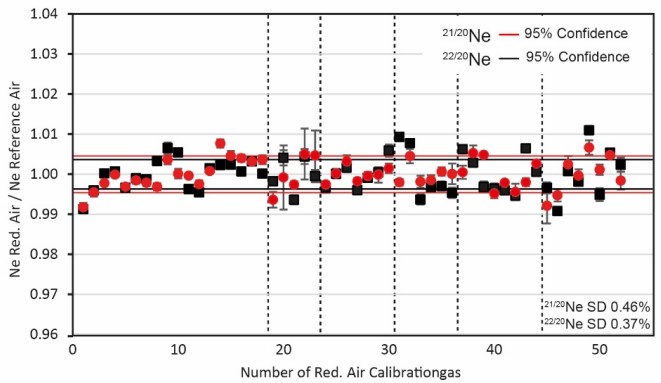


*Fig. 3: Reproducibility of standard gas 'RedAir' measurements for sample runs, during the period*
*between March 2020 and December 2020. Isotopic ratios are normalized to air for each run (mean*
*of isotope ratios obtained in run/atmospheric ratio). Stippled black lines delineate individual runs.*
*Error bars on individual data points are ±1 σ.*

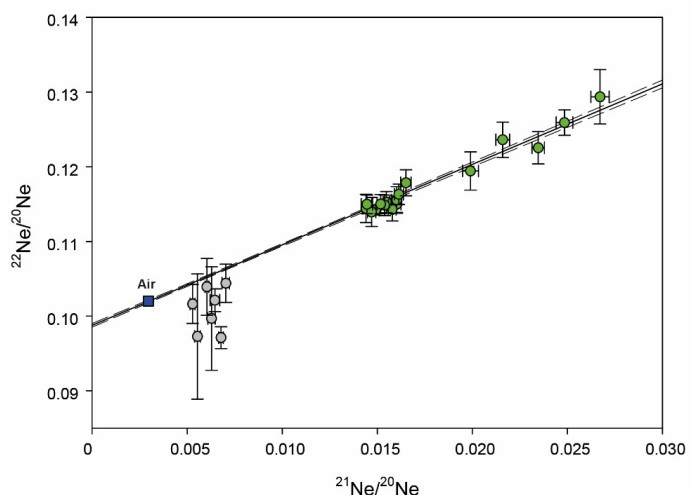


*Fig. 4: Neon-three-isotope plot for CREU-1 intercomparison material measured in Cologne. Error*
*bars are ±1 σ. The cloud of green symbols are single-step CREU extractions (100W-15min), the green*
*dots to the right of the cluster are the initial heating steps of stepwise extractions (at varying laser*
*output), grey symbols are the subsequent steps that invariably had low abundance; for details see*
*Table 1. Data of samples depicted in green are included in the regression calculation; data of the grey*
*are excluded. The slope of the regression of the data (forced through air) is 1.078±0.022 (±2σ), which*
*is indistinguishable from the published value of 1.108 ± 0.014 (±2σ; Vermeesch et al., 2015). The*
*dotted line denotes the 95% confidence interval.*

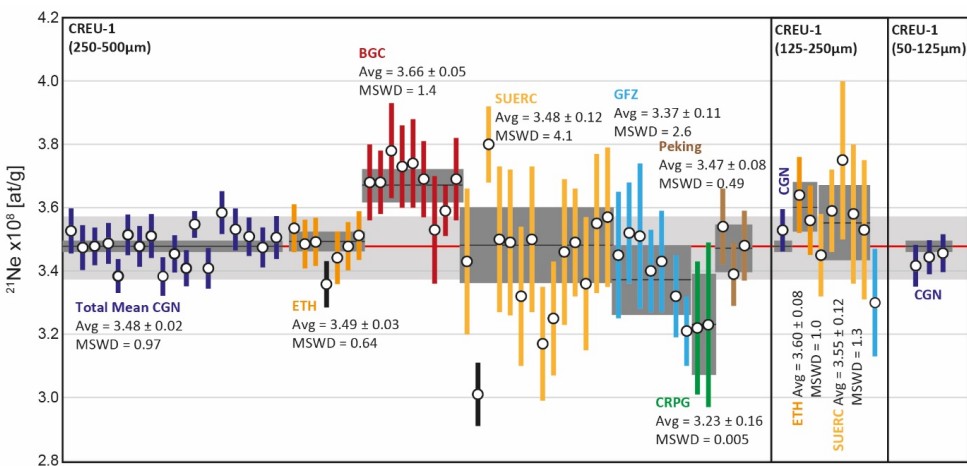


*Fig. 5: Compilation of measured CGN CREU-1 $^{21}$Ne concentrations (±2σ uncertainties), compared to*
*reported $^{21}$Ne concentrations from interlaboratory comparison from Vermeesch et al. (2015) and*
*recent data from the Peking noble gas lab from Ma et al. (2015). Black bars were considered outliers*



*by the original authors and not used for calculation of averages (Vermeesch et al., 2015). The data is*
*divided into three sections, each for a different CREU-1 grain-size analysed. The average $^{21}Ne$*
*concentration for CREU-1 of 3.48 ± 10 x $10^8$ at/g reported by Vermeesch et al. (2015) is marked as*
*light-grey band and a red line for the mean. Lab-individual error-weighted means are displayed as*
*black lines with their respective uncertainty in dark grey. The average obtained for CREU-1 at*
*Cologne (all grain-sizes, n=22) is 3.48± 0.02 x $10^8$ at/g (±2 σ; error-weighted standard deviation).*
*The MSWD values ( Mean Square of the Weighted Deviates ('reduced Chi-square', Mcintyre et al.*
*(1966)) are reported for all individual laboratory-means (Vermeesch et al., 2015; this study).*
Table 1: CREU Data

| Sample ID | Mass [g] | Extraction Power [W] | $^{20}Ne$ [$10^9$ at/g] | | | 21/20 | | | 22/20 | | | $^{21}Ne^*$ [$10^6$ at/g] | | |
|---|---|---|---|---|---|---|---|---|---|---|---|---|---|---|
| 01_CREU1 250-500µm | 0.0997 | 100 | 30.97 | ± | 0.15 | 0.01434 | ± | 0.00014 | 0.11381 | ± | 0.00129 | 352.7 | ± | 3.6 |
| 02_CREU1 250-500µm | 0.0993 | 100 | 29.81 | ± | 0.24 | 0.01461 | ± | 0.00015 | 0.11415 | ± | 0.00129 | 347.4 | ± | 3.5 |
| 03_CREU1 50-125µm | 0.1038 | 100 | 24.89 | ± | 0.15 | 0.01669 | ± | 0.00012 | 0.11646 | ± | 0.00170 | 341.7 | ± | 3.3 |
| 04_CREU1 50-125µm | 0.1319 | 100 | 25.73 | ± | 0.20 | 0.01634 | ± | 0.00018 | 0.11727 | ± | 0.00089 | 344.4 | ± | 2.7 |
| 05_CREU1 50-125µm | 0.1179 | 100 | 24.86 | ± | 0.17 | 0.01686 | ± | 0.00017 | 0.11614 | ± | 0.00130 | 345.6 | ± | 3.0 |
| 06_CREU1 125-250µm | 0.1078 | 100 | 27.00 | ± | 0.35 | 0.01603 | ± | 0.00022 | 0.11499 | ± | 0.00110 | 352.9 | ± | 3.3 |
| 07_CREU1 250-500µm | 0.1210 | 100 | 29.13 | ± | 0.36 | 0.01490 | ± | 0.00021 | 0.11417 | ± | 0.00065 | 347.8 | ± | 3.0 |
| 08_CREU1 250-500µm | 0.1105 | 100 | 30.46 | ± | 0.25 | 0.01441 | ± | 0.00018 | 0.11443 | ± | 0.00189 | 348.7 | ± | 3.2 |
| 09_CREU1 250-500µm | 0.1312 | 100 | 26.37 | ± | 0.32 | 0.01579 | ± | 0.00021 | 0.11428 | ± | 0.00153 | 338.4 | ± | 2.7 |
| 10_CREU1 250-500µm | 0.1128 | 100 | 29.92 | ± | 0.40 | 0.01470 | ± | 0.00022 | 0.11395 | ± | 0.00195 | 351.4 | ± | 3.2 |
| 11_CREU1 250-500µm | 0.1113 | 100 | 26.60 | ± | 0.33 | 0.01603 | ± | 0.00026 | 0.11556 | ± | 0.00179 | 347.8 | ± | 3.2 |
| 12_CREU1 250-500µm | 0.1053 | 100 | 30.56 | ± | 0.36 | 0.01445 | ± | 0.00030 | 0.11498 | ± | 0.00125 | 351.1 | ± | 3.5 |
| 13_CREU1 250-500µm | 0.1125 | 100 | 27.01 | ± | 0.24 | 0.01548 | ± | 0.00017 | 0.11526 | ± | 0.00143 | 338.3 | ± | 3.0 |
| 14_CREU1 250-500µm | 0.1210 | 100 | 26.21 | ± | 0.30 | 0.01614 | ± | 0.00024 | 0.11632 | ± | 0.00137 | 345.4 | ± | 3.0 |
| 15_CREU1 250-500µm | 0.1252 | 100 | 25.16 | ± | 0.19 | 0.01651 | ± | 0.00027 | 0.11786 | ± | 0.00174 | 340.9 | ± | 2.9 |
| 16_CREU1 250-500µm | 0.2063 | 100 | 28.54 | ± | 0.33 | 0.01539 | ± | 0.00020 | 0.11485 | ± | 0.00137 | 354.8 | ± | 2.1 |
| 17_CREU1 250-500µm | 0.1077 | 100 | 27.90 | ± | 0.30 | 0.01518 | ± | 0.00019 | 0.11502 | ± | 0.00129 | 340.9 | ± | 3.2 |
| 18_CREU1 250-500µm | 0.1126 | 30 | 16.44 | ± | 0.16 | 0.02160 | ± | 0.00036 | 0.12362 | ± | 0.00236 | | | |
| | 0.1126 | 50 | 8.61 | ± | 0.08 | 0.00628 | ± | 0.00019 | 0.09968 | ± | 0.00692 | | | |
| | 0.1126 | 70 | 5.71 | ± | 0.05 | 0.00528 | ± | 0.00013 | 0.10162 | ± | 0.00260 | | | |
| | 0.1126 | 100 | 3.92 | ± | 0.05 | 0.00554 | ± | 0.00015 | 0.09727 | ± | 0.00838 | 358.5 | ± | 3.4 |
| 19_CREU1 250-500µm | 0.1111 | 24 | 11.96 | ± | 0.17 | 0.02672 | ± | 0.00048 | 0.12936 | ± | 0.00364 | | | |
| | 0.1111 | 100 | 18.06 | ± | 0.23 | 0.00678 | ± | 0.00012 | 0.09711 | ± | 0.00146 | 353.2 | ± | 3.3 |
| 20_CREU1 250-500µm | 0.1301 | 24 | 14.27 | ± | 0.12 | 0.02347 | ± | 0.00033 | 0.12255 | ± | 0.00217 | | | |



| | 0.1301 | 100 | 16.51 | ± | 0.18 | 0.00646 | ± | 0.00025 | 0.10213 | ± | 0.00152 | 350.9 | ± | 3.0 |
|---|---|---|---|---|---|---|---|---|---|---|---|---|---|---|
| 21_CREU1 250-500µm | 0.1180 | 30 | 13.11 | ± | 0.15 | 0.02485 | ± | 0.00044 | 0.12592 | ± | 0.00170 | | | |
| | 0.1180 | 100 | 14.79 | ± | 0.09 | 0.00703 | ± | 0.00020 | 0.10440 | ± | 0.00256 | 347.5 | ± | 3.2 |
| 22_CREU1 250-500µm | 0.1075 | 50 | 18.42 | ± | 0.12 | 0.01991 | ± | 0.00042 | 0.11944 | ± | 0.00256 | | | |
| | 0.1075 | 100 | 12.35 | ± | 0.11 | 0.00604 | ± | 0.00010 | 0.10391 | ± | 0.00381 | 350.6 | ± | 3.4 |


## Conclusion

The performance of the set-up for Neon-isotope measurements in the new noble gas laboratory
at the University Cologne permits state-of-the art analysis of cosmogenic neon. We now regularly
perform analysis of samples for cosmogenic neon for our running projects; and are open to new
scientific cooperations.

### Author contribution:

TJD, BR, AV build-up of the noble gas system. TJD, BR performance experiments and tests. BR, TJD
manuscript writing.

### Data Availability:

The authors confirm that the data supporting the findings of this study are available within the
article.

### Acknowledgements:

The equipment for the noble gas mass spectrometry laboratory described in this paper was
funded by Deutsche Forschungsgemeinschaft (DFG) - project number 259990027 to TJD. The
performance test was conducted and funded in the framework of the Collaborative Research
Center 1211 – Earth Evolution at the Dry Limit, Deutsche Forschungsgemeinschaft (DFG) - project
number 268236062 – SFB 1211. Special thanks go to Dave Wanless for patient training and
continuing support in mastering 'Aura'.

### Declaration of interest

The authors declare that the research was conducted in the absence of any commercial or financial
relationships that could be construed as a potential conflict of interest.

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
