# Peer review of "Technical Note: Noble gas extraction procedure and performance of the 1 Cologne Helix MC Plus multi-collector noble gas mass spectrometer for 2 3"

_Geochronology, 2021_

## Referee Comment (RC1)

Review of manuscriptNr. gchron-2021-11 submitted to Geochronology: Noble gas extraction procedure and performance of the Cologne Helix MC Plus …by B. Ritter, A. Vogt and T. J. Dunai

Zürich, May 21, 2021

Rainer Wieler

The manuscript presents a new noble gas mass spectrometer system and its performance set up at the Institute of Geology and Mineralogy at the University of Cologne. The system will be used (mainly?) for the analysis of cosmogenic noble gases (primarily Ne) in terrestrial samples. The manuscript is basically well written and gives a very detailed description of the mass spectrometer and its extraction line, including a description of its operation etc. The authors also present an extensive data set on international standard quartz sample. These data document a very good reproducibility of the Ne analyses with the new Cologne instrument and also a very good agreement with data by other laboratories. In summary, this manuscript will be a welcome addition to the technical literature on noble gas mass spectrometry and deserves to be published with minor to moderate modifications as detailed in the following (numbers in the following refer to line numbers in the pdf manuscript file).

Though the manuscript in general is written in a satisfactorily style, it nevertheless requires some language polishing (including corrections of clear mistakes). I urge the senior author to keep an eye on this! In the following, I just note some of the grammatical/stylistic issues that should be considered, but this list is not exhausive!

19: automATized
24: ..is equal TO or better
36: delete "isotope geochronology"
38: delete "applying"
44: For THE evaluation of ..
49: delete one of the "atmospheric"
52: this not only holds for quartz

62ff: There has been quite some controversy recently about the 21Ne/22Ne ratio in the atmosphere. If Wielandt and Storey 2019 are mentioned here, also Saxton J.M (2020) J. Anal. Atomic Spectroscopy should be addressed. Also there is another paper on the issue by the Glasgow group (about which I am sceptical but this is my personal opinion).

section 2.2., first paragraph: refer to Fig. 1 already at the beginning of this section, when you refer to "the original …extraction ..line (now the reader has to wait until the end of the first paragraph to find this reference to the figure.

105: What is the "original" extraction and purification line. Is there any subsequent modification of this line, and who would have provided the "original"?

110: MADE of metal

127f: I had a few problems to follow this paragraph. E.g. what is a fiberlaser? or, e.g..at 131f "heating occurs with a defocussed continuous …scanning over the lids for 15 min." Perhaps it would be halpful here to start the explanations with the sample revolver (now introduced at 139-142). I even recommend to explain all this with a figure (which might be more instructive than, e.g., the lowermost panel of the present Fig.1, see below). I note that you plan to publish this new furnace elsewhere, but here it will just be very difficult to follow your description.

148 – 153, please reformulate, split long sentences.

159ff: where are all these numbers from? The manufacturer? In any case, can it be taken for granted that the isotopic composition fo Ne-Xe is exactly atmospheric, and not perhaps slightly fractionated? For example, has this been verified by comparative analyses of noble gases directly taken from air? Or can the manufacturer convincinly guarantee for a negligible isotopic fractionation?

167: It would be nice to get some additional information about the accuracy of this volume determination. At 174 we learn that the Redair pipette has a volume of ~1.5 cm3, corresponding to rougly 2 g of air. How large is the mass of the pipette and how well can then the  ~2g be measured as difference of two  weight measurements? ( I presume for the 8.7 l reservoir the problem will be much less severe?)

198: "distilled-off in disequilibrium". Explain this in more detail. In 277 we learn that He is removed from the sample gas by pumping during 5 min. Is this the same as what is said at 198? If so, how sure are you that no Ne gets lost in this process?

203: is this liquid nitrogen cooled trap filled with charcoal or another material?

Fig. 1: In my view, the large lowermost panel is unnecessary, as it does not convey any real information. As noted above, a figure of the furnace would be more instructive.

231: exlain better what a "VI" is.

293: somewhat awkward English

Fig. 2: Also this figure is not very helpful (it would in any case have to be blown up quite a bit to be readable).

255: awkward English (the lid has an opening in the lid).

261: ..transferred from the glass vials into the cup through…

277: see 198

311: please quantify: how much larger is the dispersion compared to the formal analytical uncertainties. This is a bit difficult to see in Fig. 3, as error bars are mostly not shown, and no

statement is given in the figure caption whether error bars not shown are smaller than symbol size (as I presume).

314: Is the "uncertainty of the mean" equal to the standard deviation/sqrt(n-1)? Or do you mean the standard deviation?

Fig. 5, caption: CGN? This is the first time in the manuscript this acronym is used. It must refer to the Cologne Lab, but what does it mean? Explain please (here or earlier in the text). Apart from this, your data in this figure look really nice!

357: it would be "BuilT-up" but this is not a good word here anyway, I guess.

---

## Author Response (AR1)

**Ref:    GCHRON-2021-11**

**Title:**

Technical Note: Noble gas extraction procedure and performance of the Cologne Helix MC Plus multi-collector noble gas mass spectrometer for cosmogenic neon isotope analysis

**Journal: Geochronology**

**Status:**

**Received 15 Apr 2021**

**Accepted for Review 24 Apr 2021**

**Discussion started 26 Apr 2021**

Dear Cecile Gautheron,

thank you very much for your work. In the following I will outline every change made, based on the comments of the reviewer and where appropriate provide suitable rebuttals. The line numbers we note in our attached responses refer to the revised version of our manuscript, now attached. Minor spelling and grammatical mistakes were corrected and not specifically marked in the manuscript.

Kind regards,

Benedikt Ritter

University of Cologne – Institute of Geology
* * *
Reviewer #1 (Remarks to the Author): Rainer Wieler

The manuscript presents a new noble gas mass spectrometer system and its performance set up at the Institute of Geology and Mineralogy at the University of Cologne. The system will be used (mainly?) for the analysis of cosmogenic noble gases (primarily Ne) in terrestrial samples. The manuscript is basically well written and gives a very detailed description of the mass spectrometer and its extraction line, including a description of its operation etc. The authors also present an extensive data set on international standard quartz sample. These data document a very good reproducibility of the Ne analyses with the new Cologne instrument and also a very good agreement with data by other laboratories. In summary, this manuscript will be a welcome addition to the technical literature on noble gas mass spectrometry and deserves to be published with minor to moderate modifications as detailed in the following (numbers in the following refer to line numbers in the pdf manuscript file).

Though the manuscript in general is written in a satisfactorily style, it nevertheless requires some language polishing (including corrections of clear mistakes). I urge the

senior author to keep an eye on this! In the following, I just note some of the grammatical/stylistic issues that should be considered, but this list is not exhausive!

We want to thank Prof. Rainer Wieler for his valuable and in-depth review of our manuscript, which, from our perspective, improved our manuscript significantly. In the following we outline all changes made, in response on the comments of the reviewer, where appropriate we provide suitable rebuttals.

19: automatized → corrected

24: ..is equal TO or better → corrected

36: delete "isotope geochronology" → corrected

38: delete "applying" → corrected

44: For THE evaluation of .. → corrected

49: delete one of the "atmospheric" → corrected

52: this not only holds for quartz → this is true, however, in this manuscript we focus on the analysis of Ne in quartz for cosmogenic nuclide analysis.

62ff: There has been quite some controversy recently about the 21Ne/22Ne ratio in the atmosphere. If Wielandt and Storey 2019 are mentioned here, also Saxton J.M (2020) J. Anal. Atomic Spectroscopy should be addressed. Also, there is another paper on the issue by the Glasgow group (about which I am sceptical but this is my personal opinion).

→We added a citation of Saxton (2020) and Györe et al. (2019) to the manuscript.

section 2.2., first paragraph: refer to Fig. 1 already at the beginning of this section, when you refer to "the original …extraction ..line (now the reader has to wait until the end of the first paragraph to find this reference to the figure. → corrected

105: What is the "original" extraction and purification line. Is there any subsequent modification of this line, and who would have provided the "original"?

→Corrected to "The Cologne noble gas extraction and….."

110: MADE of metal → corrected

127f: I had a few problems to follow this paragraph. E.g. what is a fiberlaser? or, e.g..at 131f "heating occurs with a defocussed continuous …scanning over the lids for 15 min." Perhaps it would be helpful here to start the explanations with the sample revolver (now introduced at 139-142). I even recommend to explain all this with a figure (which might be more instructive than, e.g., the lowermost panel of the present Fig.1, see below). I note that you plan to publish this new furnace elsewhere, but here it will just be very difficult to follow your description.

→Modified this paragraph according to the reviewer's comment, starting now with the sample revolver and then describing the laser heat extraction. Line 138-164

148 – 153, please reformulate, split long sentences. → corrected, split the long sentence.

159ff: where are all these numbers from? The manufacturer? In any case, can it be taken for granted that the isotopic composition for Ne-Xe is exactly atmospheric, and not perhaps slightly fractionated? For example, has this been verified by comparative analyses of noble gases directly taken from air? Or can the manufacturer convincingly guarantee for a negligible isotopic fractionation?

→The numbers are from the manufacturer (we now clearly state this in the manuscript). The manufacture does not guaranty the isotopic ratios of Ne, Ar, Kr and Xe. The noble gases were purified via cryogenic separation form air, we assume atmospheric composition for such gases. We have verified within the stated uncertainties this for Ne (using our 'RedAir'). The enriched Helium, the gravimetric mixture is certified by Linde via mass spectrometric analysis, the numbers we report are from Linde (DIN ISO 6141). We added this information to the manuscript. We currently do not use 'Linde' for Ne-calibrations; we use 'RedAir'. Line 172-174

167: It would be nice to get some additional information about the accuracy of this volume determination. At 174 we learn that the RedAir pipette has a volume of ~1.5 cm3, corresponding to roughly 2 g of air. How large is the mass of the pipette and how well can then the ~2g be measured as difference of two weight measurements? (I presume for the 8.7 l reservoir the problem will be much less severe?)

→The only volume that is calibrated gravimetrically is the assembly of a Swagelok SS-4H valve and a Swagelok SS-4CS-TW-50 miniature cylinder, which was used as the reference volume. All other volumes of the calibration setup (incl. pipettes and cylinders) were determined via measuring the relative pressure drop upon expansion of gases it the respective volumes. We will reword this section (to make clear that volumes are calibrated against a gravimetrically determined reference volume), to avoid future misreading. It would indeed be impractical to weigh the differences of full and empty pipettes or cylinders. We found that one mistake in the original manuscript, the uncertainty for of the pipette volume is '±0.006' (i.e. ±0.4%) rather than '±0.0006' (i.e. ± 0.04%); we will change the final manuscript accordingly. Line 180-196

198: "distilled-off in disequilibrium". Explain this in more detail. In 277 we learn that He is removed from the sample gas by pumping during 5 min. Is this the same as what is said at 198? If so, how sure are you that no Ne gets lost in this process?

→We now show the desorption characteristics of the coldtrap in a new figure showing that >99.9985% Ne is retained at 24K. Line 217

203: is this liquid nitrogen cooled trap filled with charcoal or another material?

→We do not use a liquid nitrogen cooled trap. The closed cycle refrigerated double-cold trap unit used (Janis, twin coldhead model 204) has of one bare (empty) trap and one charcoal-filled trap; the latter is currently not used for Ne separation.

Fig. 1: In my view, the large lowermost panel is unnecessary, as it does not convey any real information. As noted above, a figure of the furnace would be more instructive.

→we rather keep this panel since it helps to visualise the modular nature of the design. As mentioned, a detailed description of the furnace will follow, as soon as another development (which is close to conclusion) that relies on more demanding aspects of the furnace-design, has been published.

231: exlain better what a "VI" is.

→LabVIEW programs-subroutines are termed virtual instruments (VIs), which consists of a block diagram, a front panel and a connector pane. In principle it is just the written code/program in the LabVIEW environment. We added '(Virtual Instrument, program codes)'. Line 250

293: somewhat awkward English → corrected

Fig. 2: Also this figure is not very helpful (it would in any case have to be blown up quite a bit to be readable).

→We improved the quality and size of this image.

255: awkward English (the lid has an opening in the lid). → corrected

261: ..transferred from the glass vials into the cup through… → corrected

277: see 198 →see our reply at 198.

311: please quantify: how much larger is the dispersion compared to the formal analytical uncertainties. This is a bit difficult to see in Fig. 3, as error bars are mostly not shown, and no statement is given in the figure caption whether error bars not shown are smaller than symbol size (as I presume).

→Uncertainties are smaller than the symbol size, we added a sentence to the figure caption.

314: Is the "uncertainty of the mean" equal to the standard deviation/sqrt(n-1)? Or do you mean the standard deviation?

→We mean the standard deviation. In instances where we use standard deviation/sqrt(n-1) we specifically state that we calculated an error weighted mean.

Fig. 5, caption: CGN? This is the first time in the manuscript this acronym is used. It must refer to the Cologne Lab, but what does it mean? Explain please (here or earlier in the text).

→This is our abbreviation of our lab CGN = Cologne. We added ..CGN (Cologne) … in the figure caption.

Apart from this, your data in this figure look really nice!

357: it would be "BuilT-up" but this is not a good word here anyway, I guess.

→corrected and change to ..” development of the Cologne….”
* * *
Reviewer #2 (Remarks to the Author): Anonym

The manuscript describes the operation of a new noble gas mass spectrometer at the University of Cologne, as well as the laboratory procedures and confirmation that the performance of the equipment is on par with other laboratories. Some of the technical descriptions could benefit from more detailed explanation and clarification. The authors describe their method in great detail, I found the manuscript interesting and informative, and I recommend it be published with minor corrections.

We want to thank the anonymous reviewer for the detailed review of our manuscript and for the helpful suggestions which, from our perspective, improved our manuscript a lot. In the following we will outline every change made, based on the comments of the reviewer and where appropriate provide suitable rebuttals.

The manuscript would benefit from better proof reading. There are a few themes throughout:

- use of semi colons rather than commas (e.g., lines 21, 33, 34) and hard to read sentences (e.g., line 239) →checked and corrected accordingly throughout the manuscript

- missing gaps between number and unit (e.g., lines 133, 134, 192...) →checked and corrected accordingly throughout the manuscript

- incorrect capitalisation (e.g., lines 17, 152, 154) →checked and corrected accordingly throughout the manuscript

- Please consider accessibility with the plots. Some are hard to read because the text is too small. If you haven't already, check that some of the more colourful plots and schematics are colour-blind friendly, and consider using symbols to differentiate, rather than for example "the green cluster".

→Thank you for this valuable comment. We modified our figures. In Fig. 4 and Fig. 5, we changed the symbology.

Specific comments:

89 - 103: L1, Ax and H1 seem to be just thrown in here with no definition. Perhaps in line 85 you could define these?

→ We now added: "The central, axial module (Ax) is fixed in position, the four remaining modules (L1, L2 on the low mass side, and H1, H2 on the high mass side of Ax) can be moved.". Line 86-88

195-207: Do you have any quantitative data to support the calibration of the bare cold trap. I'd be more interested in seeing plots of how the ad/desorption varies with temperature (and that 100% of the neon is released at 80K), and what the RGA sees when you de-gas the cold trap, rather than Fig 1B. This is particularly of interest because you make comparisons with the disadvantages of a liquid nitrogen cold trap. It would be good to see the data or citations backing this up.

→We now provide a figure (Fig. 2) with the desorption characteristics of the cold trap used.

220- 222: Be very careful saying that automation helps 'prevent' oversight and negligence on the part of an operator! This view could bring in errors due to the _expectation_ that automation is infallible. Do you have safeguards in place, will you know if a automatic valve failed to open during a run? Also, later when you talk about in-house software, is this available to scrutiny?

→ We do not have specific safeguard protocols for the pneumatic valves. A failure of a valve would show up during a subsequent calibration. Spring loaded diaphragm valves (normally closed) don't fail to open and then 'decide' to work again at a later time (as is sometimes the case with gate valves or bellow sealed valves). If the pressurized air supply that powers the valves fails, we will notice immediately since no gas would be inlet into the mass spectrometer (the inlet valve is attenuated by the same pressure reservoir as are all other Fujikin valves). We write with reason 'helps to prevent' rather than 'prevents' as we concur that no system is failproof. The LabView code we wrote to integrate devices is not available for the public.

263 - The pressure being less than the gauge is capable of is good, but it does depend on where the pressure gauge is. If it right next to the turbo pump (it's hard to see from Fig. 1B but this appears to be the case), you're not measuring the pressure in the furnace, you're measuring the pressure at the pump. It that's the case the blank is more important to report here than the pressure.

→We report the pressure since we describe the protocol. We decide to start a sample run after sample reload when the pressure reading is low (i.e., in the $10^{-9}$mbar range), then we measure blanks. We provide the information on typical blank levels (~0.3 fmol Neon) before, in the section where we describe the laser furnace.

275 - define "hot" - what temperature are you running the hot getter at?

→We originally did not provide the temperature, since we cannot measure it (the cartridges are internally heated and have no thermocouple). We will now report the heating current (1.6 A) of the SAES cartridge and the estimated temperature (~300ºC) derived from the corresponding diagram in the SAES data sheet.

302: You calibrate with RedAir once per day to check mass discrimination, sensitivity and multiplier vs faraday gain. Is this enough? Do you observe changes (particularly multiplier vs faraday) over the course of the day as you run experiments? I'd expect at the start of a day the mass spec has had time to 'reset' overnight, so a calibration run every morning

might be broadly consistent with the previous day, but over the course of the day there could be lots of sensitivity changes (especially with large signals). Also, you say "at least" once per day. Are there reasons why you might do more than one, or is there no set pattern?

→After changing the isotope system (from He to Ne), reloading of samples or after starting a new measurement series after the machine was idle for some time, we measure several (>5) RedAirs to 'wake-up' the multipliers and asses their stability, and that of the mass-spectrometer. After this initial wake-up protocol we do not observe changes in the sensitivity over the course of a day, or between days. Thus, the default is to run one calibration per day, as first measurement of a given day. To date all samples had Ne-abundances small than the calibration gas, thus do not modify multiplier sensitivity. In cases where the operator judges that any sample's result is in any way unusual, they can perform (an) additional RedAir calibration(s) at any day, to ensure that the extraction sequence and the mass spectrometer is running normally (which is, so far, always the case). Running an additional calibration at the end of a given day comes at little time expense, since the gases are purified automatically, and the gas inlet can be performed remotely (via. TeamViewer).

322 - Figure 3. I cannot tell if this is just me seeing a pattern in the data, but does the dispersion increase with time? It would also be helpful, for accessibility, to have the symbols in the key, not just the colours.

→Similar to the reviewer we can't discern a significant trend in the dispersion with the current data; concerning accessibility Fig. 3 will be modified following the reviewer's suggestions.

Typos etc (not a complete list):

Line 13 - The opening line seems a bit cluncky - use "dedicated to" rather than "dedicated for"? (first few sentences could do with reworking) →corrected

17 - mass spectrometer, not Mass-spectrometer →corrected

19 - automated would read better than automized (section 2.3 is subtitled automation) →corrected

62 - 65 - This sentence is hard to read. Maybe "Common isobaric..... are at: m/e = 20 (interferences on $20Ne^+$ are $40Ar_2^+$, $H^{19}F^+$, $H_2^{18}O^+$), m/e=21 (interferences on $21Ne^+$ are ....) ....etc" →corrected

85 - "five CFM modules" or "five CFMs", not "five CFMs modules" →corrected

110 - made of metal →corrected

198 ion pump not ion-pump →checked and corrected throughout the manuscript

152 - 5*10 - use 'x' instead of * and 156.6 not 156,6 →corrected

171 - gases not gasses →corrected

189 - the starcell is referred to here as an iongetter (should be ion getter?) pump but in 193 as an ion pump. Maybe just use ion pump here →corrected

280 - resulting rather than ensuing? →Changed to "Subsequently, Ne gas …"
* * *
**Editor Comment: Cecile Gautheron 09.07.2021**

Dear Ritter and co-authors,

Thanks for the reply to the reviewers' comments that are well addressed. However, I have some additional comments to the reviewers, as I believe that some other small details need to be addressed.

Firstly, please define the name of redair, sputnik.

Fig. 1 needs more symbols description in order to be understood. Could you please define all the symbols for the different units of the line?

I also suggest presenting Fig. 2 in a different shape, as in the present form it is too small.

About Fig. 3: a description the origin of some data with large error bars will help the reader.

Finally, Fig 4 and 5 should be describe in detail in the text and called that is not the case in the present form. Those two figures are not used in the present text, which is a shame considering the data quality. For Fig. 5, the name of the different labs should be given (ETH, BGC, SUERC, GFZ, Peking, CRPG) even if most of the data have been taken from Vermeesch et al. 2015.

I am looking forward reading the corrected version

Sincerely

Cécile Gautheron

We want to thank the editor Cecile Gautheron for the additional comments and suggestions to our manuscript. In the following we will outline every change made, based on the comments of the editor and where appropriate provide suitable rebuttals.

Firstly, please define the name of redair, sputnik.

➔ As stated in the manuscript Line 177-178 (revised version) ".  'Air' is a reservoir of air at atmospheric pressure and 'RedAir' a reservoir of air at reduced pressure." RedAir = Reduced Air; New in text: '(lab-name 'RedAir' is the abbreviation of that fact).'

➔ 'Sputnik' is our name for the configuration of the clean-up module, which bears resemblance to the 'Sputnik' satellite. New explanation in text: ('Sputnik', lab-name, referring to the shape and protrusions of the central manifold and its faint resemblance to the first satellite)

The symbols used in Fig. 1 are standard symbols in Vacuum technology, we now provide explanation for all symbols.

Fig.2: Fig.2 in the original manuscript is now figure 3 in the revised version. We improved the quality of the figure, so that all details can be seen.

Fig. 3: Information to data with larger uncertainties: Fig. 3 (former Fig. 2 in original manuscript) displays all RedAir calibrations measured during the period between March 2020 and December 2020. We added to the manuscript: 'The second measurement period, with the increased uncertainties of the $^{21}Ne/^{20}Ne$ ratios, was performed after an extended period of development work for other noble gas isotopes.' Line 330-332

Fig. 4 and 5.: We incorporated the Figures now in the text of the manuscript. Explanations for the abbreviations of the individual noble gas labs are now provided in the figure caption.

➔" Derived $^{21/20}Ne$ and $^{22/20}Ne$ ratios of 22 aliquots of CREU-1, including five power step extractions (Table 1), reveal a spallation line of 1.078 ± 0.022 (±2σ), which is indistinguishable from the published value of 1.108 ± 0.014 (±2σ; Vermeesch et al., 2015, Fig. 5). The calculated cosmogenic $^{21}Ne$ abundances from 22 aliquots of CREU-1 (Table 1) all agree within 2σ with their arithmetic mean (348 ± 10 * $10^6$ atoms/g; ±2σ); thus, we may calculate an error-weighted mean: 348 ± 2 * $10^6$ atoms/g (±2σ), which is indistinguishable from the published value (348 ± 10 * $10^6$ atoms/g; Vermeesch et al., 2015, see Fig. 6)." Line 333-339

Figure Caption 6 (former Fig.5): We added "*CGN = University of Cologne, ETH = Eidgenössische Technische Hochschule Zürich, BGC = Berkeley Geochronology Center, SUERC = Scottish Universities Environmental Research Centre Glasgow, CRPG = Centre de Recherches Pétrographiques et Géochimiques Nancy, GFZ = Deutsches GeoForschungsZentrum Potsdam.*"

---

## Author Response (AR2)

**Ref:    GCHRON-2021-11**

**Title:**

Technical Note: Noble gas extraction procedure and performance of the Cologne Helix MC Plus multi-collector noble gas mass spectrometer for cosmogenic neon isotope analysis

**Journal: Geochronology**

**Status:**

**Received 15 Apr 2021**

**Accepted for Review 24 Apr 2021**

**Discussion started 26 Apr 2021**

**Review Response 12 July 2021**

**Additional Review by Editor 15 July 2021**

Dear Cecile Gautheron,

thank you very much for your work. In the following we will outline every change made, based on the comments of your review and where appropriate provide suitable rebuttals. The line numbers we note in our attached responses refer to the revised version of our manuscript, now attached. Changes according to this review are marked in light blue. Changes based on the first review are marked in green.

Kind regards,

Benedikt Ritter

University of Cologne – Institute of Geology
* * *
Dear Ritter and co-authors,

Thank you for the corrected version which answers the different reviews. The text and figures are clearer and more informative; however, some small adjustment can be made. Additional details on Figures will increase the understanding on the procedure and results. Also, some small typo problems are still present in this version (see below).

Some explanations on Fig 3, 4 and 5 are still missing and link to the text is some time very poor, so please add more details on them:

-Fig 3 that is not described properly and links with Fig 1 are not made enough.

→We added additional information of Fig. 3 in the text and added additional figure reference in the manuscript. In general, Fig. 3 display just the user program interface. In

the text we wrote "For handling, a structured user program interface was designed (Fig. 3), which provides the user with information about all parameters, total duration, and additionally logs every extraction step."

→We added several figure references for Fig. 1 in the manuscript where appropriate.

Please describe what is M1, M2, … M5.

→We extended figure caption 3 with the following information: ". M1-M5 indicate the different modules of the extraction line. Valve numbers (1-10,20-31, M, T, I) are coloured depending on the current state (open or close)."

Are the number next to the valve the valve number? If yes, please add this information to Fig 1, and indicate the meaning in the legend. The red or green valves are for close and open valves? Please describe the content of the figure to help the reader to understand how your lab is functioning

→ see Fig. 3 caption. For the general information about the extraction line, it is not necessary to add the valve numbers to Fig. 1. We think that it would overload the Fig. 1. The requested information is found in Fig. 3.  The numbering of the valves is not required to understand how our lab is functioning, we do not refer to the numbers when we explain the functionality of the extraction line. Anybody can copy the line without knowing how we call the valves. We now state in the caption of Fig. 3 that open valves are depicted in green and closed ones are in red. Note: this Fig. 3 is a snapshot; the status of the valves changes during operation.

- Fig 4: please add a space between calibration and gas →corrected

I am not sure to understand the sentence of line 330-332 "the second measurement period…" How does the fact that you had a period where you developed the other noble gases change the neon data? Please add more justification and explanation with the different dataset. Please define when was the first and second measurement period? You did not explain, why some value present larger error bars than other. Please be more specific.

→ We now add "The larger errors of the $^{21}Ne/^{20}Ne$ -ratios of the second run may be due to the fact that prior to that run a longer development period of other noble gas species, and other sample materials, was conducted. During developmental work on a noble gas line, particularly when other gas species are analysed, the residual gas composition in the extraction line and in the mass-spectrometer may change. The latter may affect the response/stability of multipliers ($^{21}Ne$ is the only isotope we measure on the multipliers; thus, it is the $^{21}Ne/^{20}Ne$ that shows the higher variability)." in the figure caption. The measurement periods are separated by the stippled vertical lines, this information is provided in the caption of Fig. 4 (`Stippled black lines delineate individual runs.`). The number of the measurement periods increases from left to right, with the increasing number of calibrations.

- Fig. 5: neon isotopic ratio. → We do not understand the intention of this comment. The description of the plot as "neon three-isotope plot" is also used by Vermeesch et al. 2015

Please explain better what the initial heating steps are? you mean the first 1 to 3 steps (green dots) and the steps 4 to 5 (grey rectangles) are the subsequent steps. Be more specific on how the distinction is done?

→We stated in the figure caption (Fig. 5): "The cloud of green symbols displays single-step CREU extractions (100 W-15 min), the green dots to the right of the cluster are the initial heating steps of stepwise extractions (at varying laser output), grey rectangles are the subsequent steps that invariably had low abundance; for details see Table 1." The initial (first) heating/power step (see Table 1) will extract the majority of the neon gas from CREU-1. Subsequent heating/power steps will extract the remainder of neon in the sample or CREU-1 and will plot due to the low abundance to the left close to the air value on the neon-three-isotope plot. We added to figure caption (Fig. 5): "...the green dots to the right of the cluster are the initial heating (first extraction of a sample) steps of stepwise extractions..."Line 357-358.

Small other typo problems:

Please unify the writing of the neon isotopic ratio in the text ($21Ne/22Ne$ and not $21/22Ne$), figures and table (use $21Ne/20Ne$ and not $21/20$ etc) as the different notations are used. → corrected throughout the manuscript, figures and table

Line 280: please change $5*10-9$ by $5x10-9$→corrected

Same comment in line 339, 340, 341, 366, 369 →corrected

Line 241 put the 6 of $10^6$ in index →error not found, however, we checked the entire manuscript for this problem

Ad GCN also in fig 3 and 4 → added

Table 1: please explain what is $21Ne^*$ (the asterix is referring to what?) →modified to $^{21}Ne\ ^*cos$, for the cosmogenic $^{21}Ne$

In the acknowledgement, you can thank the reviewers → We added: "Furthermore, we want to thank Rainer Wieler and one anonymous reviewer for their constructive feedback on the submitted manuscript."

References: please add the DOI number to all references (when possible). Be careful with the writing of isotopes and molecule to put the associated symbol or number in index → added